# Electrolyte droplet spraying in H₂ bubbles during water electrolysis under normal and microgravity conditions

Aleksandr Bashkatov [1,2,3] ✉, Florian Bürkle [4], Çayan Demirkır [2], Wei Ding [1], Vatsal Sanjay [2], Alexander Babich [1], Xuegeng Yang [1], Gerd Mutschke [1], Jürgen Czarske [4], Detlef Lohse [2,5], Dominik Krug [2,3], Lars Büttner [4] & Kerstin Eckert [1,6] ✉

Electrolytically generated gas bubbles can significantly hamper the overall electrolysis efficiency. Therefore it is crucial to understand their dynamics in order to optimise water electrolyzer systems. Herein, we elucidate a distinct transport mechanism whereby electrolyte droplets are sprayed into H₂ bubbles. These droplets arise from the fragmentation of the Worthington jet, which is engendered by the coalescence with microbubbles. The robustness of this phenomenon is corroborated under both normal and microgravity conditions. Reminiscent of bursting bubbles on a liquid-gas interface, electrolyte spraying results in a flow inside the bubble. This flow couples, in an intriguing way, with the thermocapillary convection at the bubble's surface, clearly underlining the high interfacial mobility. In the case of electrode-attached bubbles, the sprayed droplets form electrolyte puddles affecting the dynamics near the three-phase contact line and favoring bubble detachment from the electrode. The results of this work unravel important insights into the physico-chemical aspects of electrolytic gas bubbles, integral for optimizing gas-evolving electrochemical systems.

The growth of gas bubbles abounds in nature and has various engineering applications[1] and is reflected in natural phenomena. Some of them, featuring rapid dynamics, are sonochemistry[2] and sonoluminescence[3], cavitation[4], the evolution of $CO_2$ bubbles in sparkling drinks[5], and the bursting bubbles at the oceans surface[6,7]. The latter contributes significantly to atmospheric aerosol generation[8] via two different mechanisms: the disintegration of a thin liquid film between the bubble and gas interface at the onset of bursting[9–11], and by an inertia-driven liquid jet–referred to as Worthington jet fragmenting into multiple droplets[12,13], where the mechanism is either end-

pinching[14–17] or Rayleigh–Plateau instability[8,18]. Beyond aerosol generation[19], these jets are responsible for contaminant dispersion[20,21]. Additionally, they result in surface erosion and deformation through the impact of droplets on solid[22] and liquid surfaces[23], respectively.

A related problem also occurs in electrolysis, where the coalescence of hydrogen or oxygen bubbles can be approximated to bursting events at a liquid-gas interface. This is a particularly interesting problem of high practical relevance due to the prominent role of hydrogen produced via water electrolysis as an energy carrier, fuel, and feedstock for chemical and steel industries[24]. Alkaline water

[1]Institute of Fluid Dynamics, Helmholtz-Zentrum Dresden-Rossendorf, Bautzner Landstrasse 400, 01328 Dresden, Germany. [2]Physics of Fluids Group, Max Planck Center for Complex Fluid Dynamics and J. M. Burgers Centre for Fluid Dynamics, University of Twente, P.O. Box 217, 7500AE Enschede, Netherlands. [3]Institute of Aerodynamics, RWTH Aachen University, Wüllnerstraße 5a, 52062 Aachen, Germany. [4]Laboratory for Measurement and Sensor System Techniques, Faculty of Electrical and Computer Engineering, Technische Universität Dresden, Helmholtzstr. 18, 01069 Dresden, Germany. [5]Max Planck Institute for Dynamics and Self-Organization, Am Fassberg 17, 37077 Göttingen, Germany. [6]Institute of Process Engineering and Environmental Technology, Technische Universität Dresden, 01062 Dresden, Germany. ✉e-mail: a.bashkatov@hzdr.de; k.eckert@hzdr.de

electrolysis is still the most mature technology, albeit suffering from inadequate efficiency when operated at high current densities. A considerable part of the losses can be attributed to the formation of $H_2$ and $O_2$ bubbles, present at the electrodes and in the bulk. These bubbles mask the active area of the electrodes, reduce the number of nucleation sites, and raise ohmic cell resistance[25,26]. Thus, enhanced removal of gas bubbles, inherently requiring a better understanding of their growth and departure, will promote continuous catalytic activity[27] and benefit further optimization of the system's energy efficiency[28].

The dynamics of electrolytic bubbles have been extensively studied in the last few decades[25,29] to uncover the growth laws controlled by either the diffusion of dissolved hydrogen[30,31] or direct injection of the gas at the bubble foot[32]; mass transfer and associated limitations[31,33]; interactions between neighboring bubbles[27,34]; the impact of the electrolyte composition[35], also in the presence of surfactants[36]; the force balance governing the bubble departure[34,37] and finally the impact of bubbles on the cell overpotentials[25,30]. Only recently, the soluto- and thermocapillary Marangoni[35,38–40] force and an electric[32,37] force caused by charge adsorption, which had not been considered before, have been uncovered and quantified. Furthermore, it has been discovered that $H_2$ bubbles on microelectrodes do not necessarily adhere to the surface. Instead, they might reside atop a "carpet" of microbubbles and grow via intensive coalescence with this bed of tiny precursors[32]. However, the full implications of such rapid coalescence events in water electrolysis remain elusive–an area ripe for further inquiry. Several lingering questions are yet to be addressed: What are the main features of the coalescence in the confined geometry, set by $H_2$ bubble, carpet and electrode, and how do they interact with the Marangoni flow at the bubble surface? Under what conditions does electrolytic bubble coalescence lead to droplet and spray formation? Does this affect the contact line and potentially the detachment of the electrode-attached bubble?

In the present work, we address these open questions by combining experiments on the coalescence-driven dynamics of $H_2$ bubbles, focusing on the interior of the bubbles under both terrestrial and microgravity environments, alongside tailored direct numerical simulations.

## Results

### Electrolyte spraying

The main phenomenon under study, spray formation inside a $H_2$ bubble during water electrolysis, is presented in Fig. 1. This observation was made under microgravity conditions provided by parabolic flights of an Airbus A310[41]. The snapshot at $t = 0$ shown in Fig. 1a documents the time instant when the bubble sits at the electrode surface, blocking most of its active area, hindering the reaction and hydrogen production rates. Figure 1b, c zooms into the central and lower segments of the bubble, respectively, at various time points leading up to its departure. Soon after $t = 0$, the bubble begins a lateral shift to the right driven by residual gravitational forces releasing the electrode and enabling the formation of a dense carpet of microbubbles. As a result, the primary bubble continuously coalescences with these microbubbles emerging on a time scale of $\mathcal{O}(\mu s)$. The successive images document an emerging flow consisting of electrolyte droplets, which is initiated soon after the onset of coalescence events and ascends from the base of the bubble toward its apex. These droplets become noticeable at $t = 0.16$ s, with their population density peaking at $t = 1.08$ s and declining by $t = 2.92$ s. The gradual widening of the gap between electrode and bubble interface over time has two different effects: (i) It enhances the electrochemical reaction (by increasing the electric current, see Supplementary Fig. 1), thereby

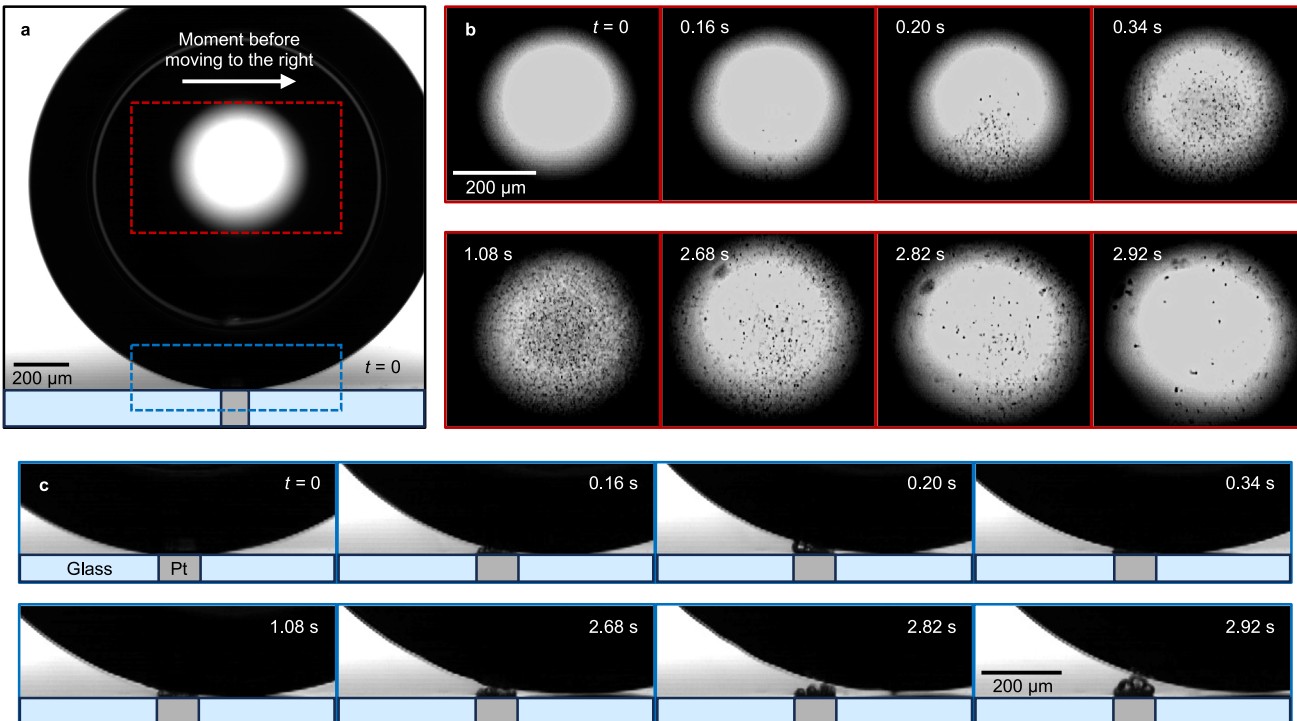

**Fig. 1 | Electrolyte spraying in a microgravity environment.** A series of shadowgraphs documenting the stream of electrolyte microdroplets inside a $H_2$ bubble ($R_b = 902\,\mu m$ at the departure) during the late phase of its evolution in a micro-$g$ environment. At $t = 0$ in (**a**) the bubble sits at the electrode. The successive images in (**b**) and (**c**), zooming into central and lower segments of the bubble, shown by red and blue rectangles in (**a**), demonstrate the emerging flow of electrolyte microdroplets, initiated soon after the onset of lateral motion to the right followed by the intensive coalescence events. The $H_2$ bubble is produced during water electrolysis at 100 μm Pt microelectrode at −4 V (vs. Pt wire) in 0.5 mol $L^{-1}$ $H_2SO_4$. The image recording was performed with a frame rate of 50 Hz. For further details, we refer to Supplementary Movie 1.

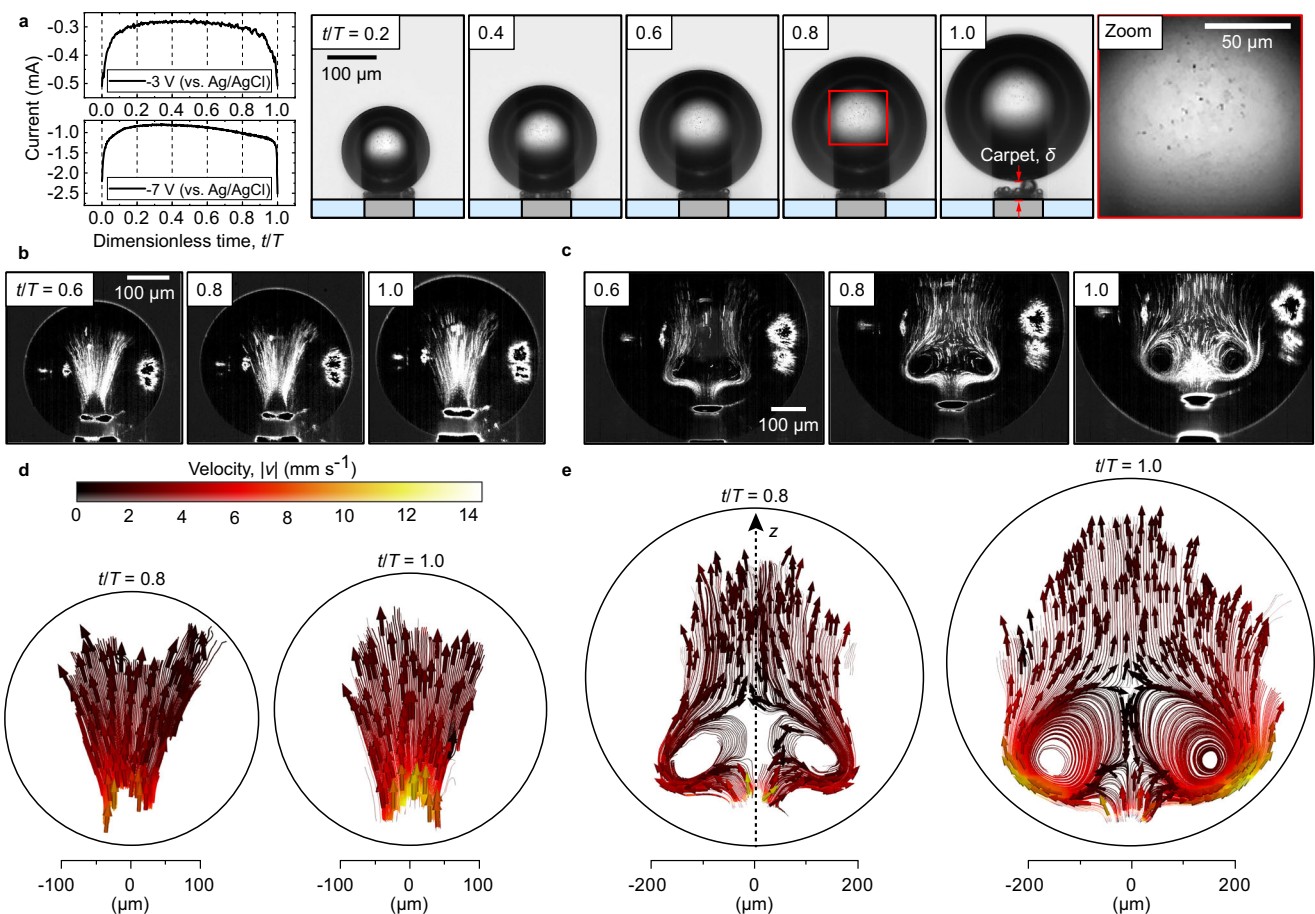

**Fig. 2 | Flow structure within H₂ bubble residing atop a carpet of microbubbles.** The dynamics of H₂ bubble presented in terms of (**a**) electric current at −3 (top) and −7 V (bottom), supplemented with shadowgraphs throughout its evolution at −3 V. The rightmost image zooms into the middle part of the bubble at $t/T = 0.8$, shown by the red rectangle. **b**, **c** Snapshots highlighting the streaklines of the droplets over $\Delta t = 25$ ms. **d**, **e** The streamlines of the averaged drop velocity field.

Before velocity calculations, optical distortions (aberration) caused by the curvature of the bubble are analytically corrected (see Methods and Supplementary Note 2). The measurements in (**b–d**) and (**c–e**) performed in 0.1 mol L⁻¹ H₂SO₄ at −3 and −7 V vs. Ag/AgCl, respectively. For further details, we refer to Supplementary Movies 2 to 4. Source data are provided as a Source Data file.

elevating H₂ production and bubble–carpet coalescence rates, and (ii) it leads to the generation of larger pre-coalescence bubbles, in turn decreasing the frequency of the coalescence events and, subsequently, droplet injections into the main bubble. The competition between these two effects establishes an optimal carpet thickness at which the coalescence rate has its maximum. Beyond this distance, the droplet population significantly reduces, as evidenced at $t = 2.92$ s. The droplet radii remain approximately constant at $1.8 \pm 0.8$ μm during most of the coalescence phase, increasing to about $3.1 \pm 1.3$ μm only just before bubble departure, when the gap between bubble interface and the electrode is at its maximum.

**Impact of Marangoni convection on internal flow structure**

In the following, we demonstrate how the phenomenon manifests itself under normal gravity conditions. Figure 2 illustrates (a) the electric current $I$ at − 3 and − 7 V, and shadowgraphs along bubble evolution at −3 V in 0.1 mol L⁻¹ H₂SO₄. In detail, a single primary bubble forms via coalescence shortly after nucleation at $t/T = 0$ from many nano- and micrometer bubbles[32]. It continues to grow through rapid $\mathcal{O}$(μs) coalescence with the carpet of microbubbles beneath. The evolution ends with the bubble departure at $t/T = 1$ when buoyancy overcomes downward forces[37]. $T$ is the bubble lifetime. $I(t)$ reflects variations in ohmic resistance due to bubble size and position relative to the electrode, peaking between the departure and nucleation of the next bubble.

In analogy to Fig. 1, numerous electrolyte droplets are injected during the coalescence events, as seen in the last image of Fig. 2a, which focuses on the central segment of the bubble at $t/T = 0.8$. The snapshots in (b,c) highlight the streaklines of the droplets over $\Delta t = 25$ ms, emerging at the bubble-carpet interface and moving towards the bubble apex with the velocities plotted in (d,e).

The flow in Fig. 2b–e develops continuously throughout the bubble evolution, along with and in response to the growing carpet thickness[32,35] and hence elevated current, reaching velocities of up to 14 mm s⁻¹ at $t/T = 1.0$. High-speed recordings at 600 and 720 kHz (Supplementary Note 3 and Supplementary Fig. 4) reveal that some droplets are injected at velocities up to 15.8 m s⁻¹, i.e. three orders of magnitude higher. These rare events, resulting in larger droplets, occur around the bubble's departure when the carpet thickness is at its maximum, approximately between $\delta = 16$ μm and $\delta = 43$ μm (see Fig. 2a), but are not observed during the earlier stages of the bubble's evolution when $\delta < 16$ μm.

At a substantially larger electric current (see Fig. 2a), the flow is altered by the presence of a vortical structure, see a transition from a fireworks-like shape (b,d) at −3 V to a vortex-like shape at −7 V (c,e). Meanwhile, the flow at the base of the injection remains similar. At lower potential, the flow expands away from the injection source, while at higher potential, the droplets are carried away from the injection area and ascend along the bubble-electrolyte interface. In the latter, some droplets enclosing the vortex are carried back toward the

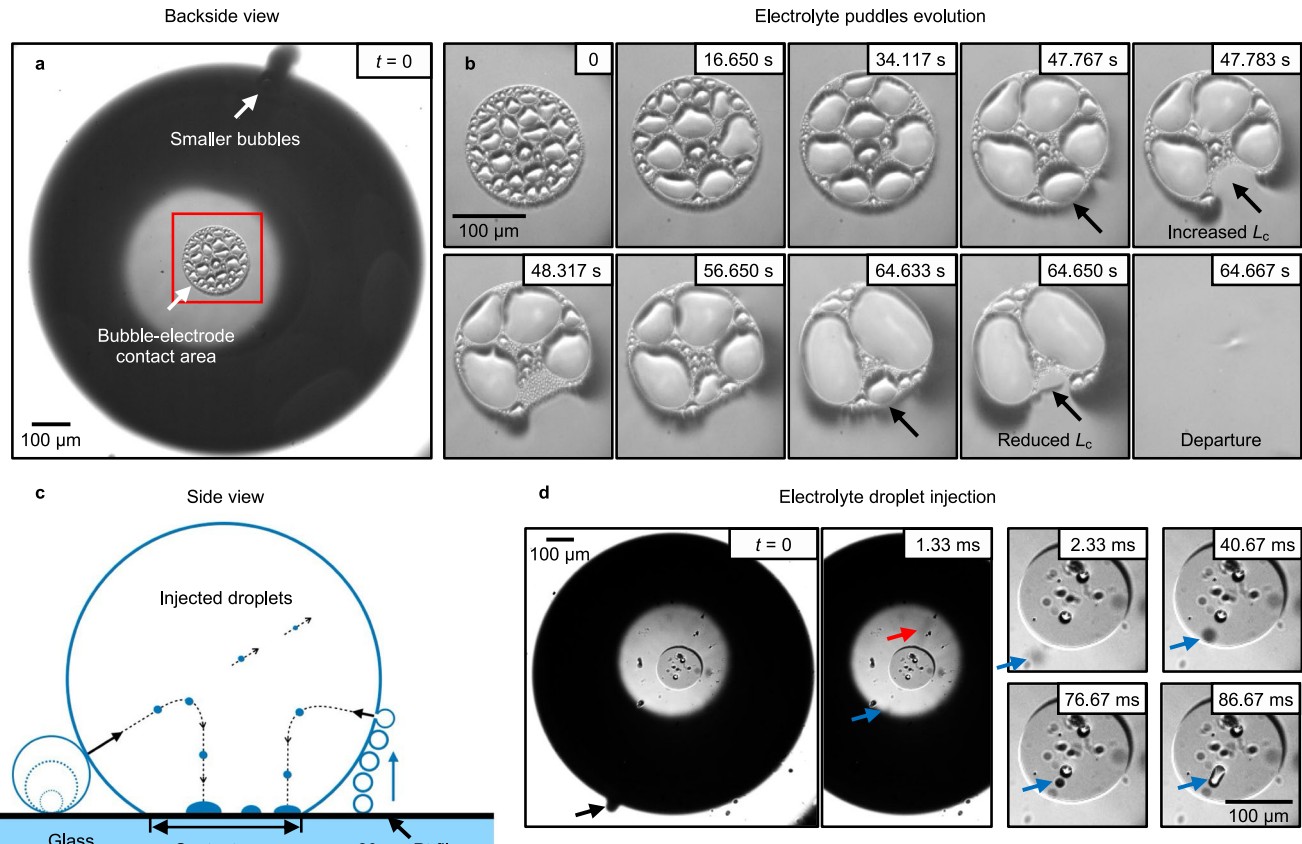

**Fig. 3 | Electrolyte puddle formation at bubble-electrode interface.**
**a**–**d** Backside views from underneath the electrode of the growing $H_2$ bubble attached to the transparent planar Pt electrode. **b** Zooms into the bubble-electrode contact area shown by the red rectangle in (**a**), demonstrating the development of electrolyte puddles throughout the bubble evolution. **c, d** Schematic and shadowgraphs illustrating the injection of microdroplets upon coalescence events followed by their sedimentation at the contact area. The measurements were carried out at a current density of $-50$ A m$^{-2}$ in 0.1 mol L$^{-1}$ HClO$_4$. The image recording in (**a**, **b**) and (**d**) was performed at frame rates of 60 Hz and 3000 Hz, respectively. For further details, we refer to Supplementary Movies 5 and 6.

electrode. In both cases, the velocity gradually decays with distance. Indeed, the velocity of injected droplets is expected to decay exponentially over time due to viscous drag: $v_d(t) = v_0 \cdot \exp(-t/\tau_d)$ (Supplementary Fig. 5). Here, $\tau_d = \frac{m_d}{6 \cdot \pi \cdot \mu_{H2} \cdot R_d}$ and $R_d$ is the radius of the droplet, $m_d$ is the mass of the droplet, $v_0$ is the initial velocity and $\mu_{H2}$ is the dynamic viscosity of $H_2$. For example, a droplet with $R_d = 1\,\mu m$ (implies $\tau_d \approx 25\,\mu s$) and initial velocity $v_0 = 5$ m s$^{-1}$ slows to $10^{-2}$ m s$^{-1}$ in just 160 μs by traveling 124 μm. At $-7$ V, droplets near the $z$-symmetry line (see Fig. 2e) are dragged into the downward flow stream, enclosing the vortex.

The flow transition observed between $-3$ and $-7$ V is due to Marangoni convection around an electrogenerated gas bubble existing at its outer interface. This convection originates from a gradient of surface tension caused by thermo- and/or solutocapillary effects[35,38,39], creating a shear stress imbalance that moves the liquid-gas interface. The resulting flow is directed alongside the electrolyte-gas interface from small to large values of surface tension, i.e., from the bottom to the top of the bubble. These effects are localized at the foot of the bubble and are consistent with the position of the vortex ring in Fig. 2e. Thermal Marangoni forces are driven by Joule heating from locally high current density ($j$) at the wetted part of the electrode and scale (via Ohm's law) with $j^2$, while solutal Marangoni forces arise from electrolyte depletion at the electrode and depend linearly on $j$. At higher potentials, as in the present study, the Marangoni convection is mainly driven by thermal effects[35], with temperature rising up to 14 K[39]. The velocity magnitude scales with the electric current[38] and may reach about 10 mm s$^{-1}$ at $-2.2$ mA and 47 mm s$^{-1}$ at $-4.8$ mA in 0.5 mol L$^{-1}$ $H_2SO_4$[42]. This concludes that the pronounced variance in flow structure between $-3$ and $-7$ V originates from the substantial difference in electric current magnitude and, consequently, the Marangoni convection. Thus, reminiscent of evaporating droplets[43] or rising bubbles[44], Fig. 2 demonstrates that Marangoni convection at the electrolyte-gas interface drives internal flow in electrogenerated gas bubbles, directing and accelerating injected microdroplets. This also indicates that the gas-electrolyte interface is mobile, though the mechanism behind preferential ion adsorption and its effects remain unclear.

## Electrolyte puddles at the bubble-electrode interface

Another intriguing outcome of the spraying, shown in Fig. 3, is the formation of electrolyte fractions within an electrode-attached and growing $H_2$ bubble, specifically at the contact area with the electrode surface[34,36]. Figure 3a, b, d documents the views from below a transparent planar electrode (20 nm of Pt). The snapshots in Fig. 3b zoom in on the contact patch (area marked by the red rectangle in Fig. 3a), which is seen to feature sessile electrolyte droplets inside the gas phase, that expands throughout the bubble evolution[34]. The bubble grows mainly due to diffusion of the dissolved gas but also via coalescence with the neighboring bubbles. Here, the smaller bubbles nucleate below the equator of the primary bubble and quickly detach, see a plume of out-of-focus small bubbles in Fig. 3a, likely due to the altered morphology/wettability of a tiny cavity they nucleated at. Consequently, upon reaching the gas-liquid interface of the larger bubble, coalescence occurs between the two, see schematic in Fig. 3c.

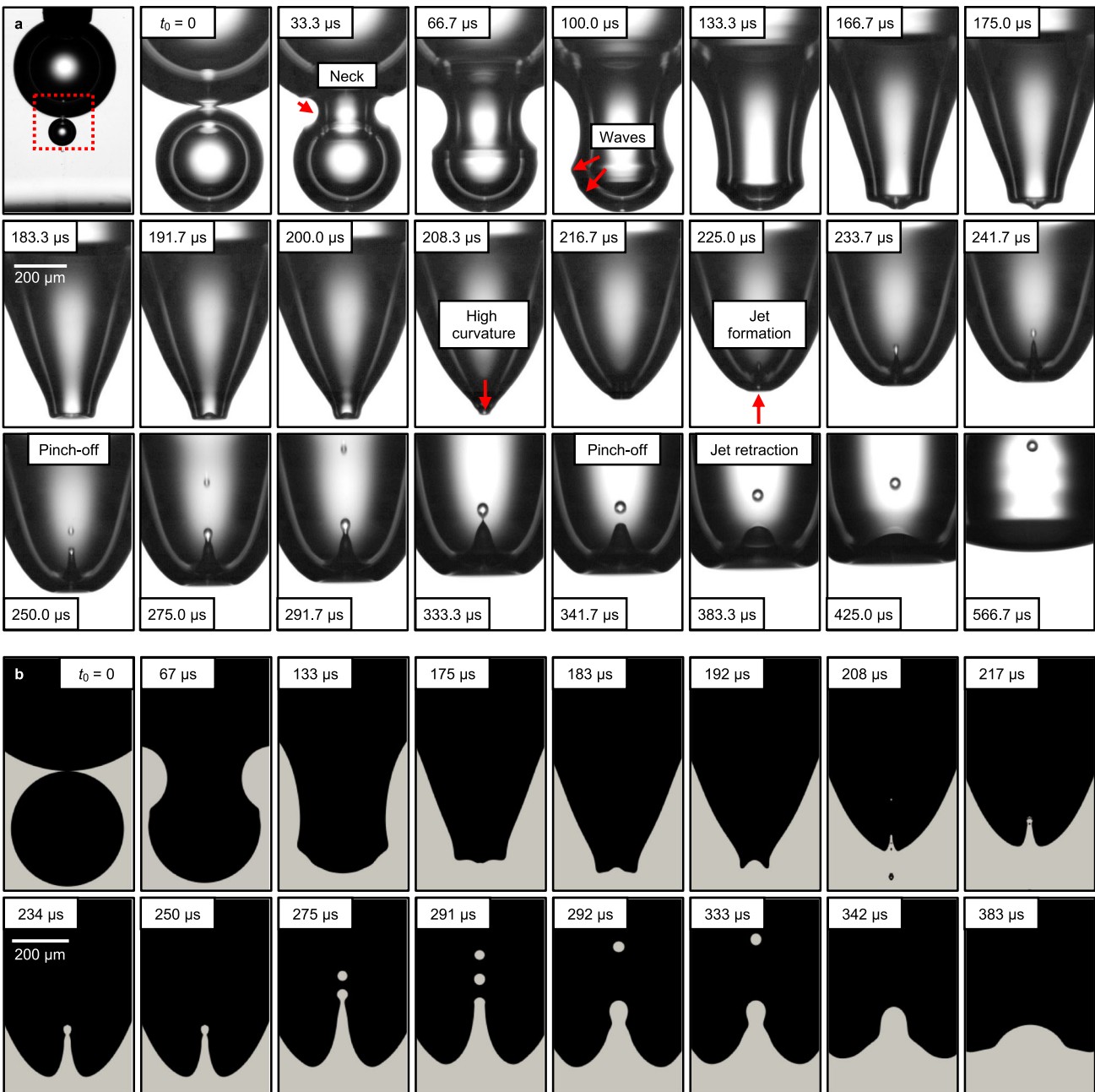

**Fig. 4 | Droplet ejection mechanism.** Coalescence of two $H_2$ bubbles of unequal size, shown by snapshots from (**a**) an experiment and (**b**) a numerical simulation. In the experiment, both bubbles were produced during electrolysis in 0.5 mol $L^{-1}$ $H_2SO_4$. While the bigger bubble is pinned to a blunt needle, the smaller bubble rises from the electrode until the coalescence begins at $t_0 = 0$. The coalescence process is accompanied by the injection of two droplets after their consecutive separation from the jet. The first snapshot in (**a**) (not to scale) demonstrates the configuration, marking the region of interest with a red rectangle. The image recording was performed at 120 kHz. For further details, we refer to Supplementary Movie 7.

Figure 3d details the injection of at least two microdroplets, marked by red and blue arrows, following the coalescence event between the primary and smaller bubble (black arrow at $t = 0$). The first droplet (red arrow) moves with a much faster velocity, likely shooting through the gas-electrolyte interface on the opposite side. In contrast, the second droplet, about $r = 4$ μm (blue arrow), slows down quickly due to Stokes's drag (Supplementary Note 4 and Supplementary Fig. 5) and falls, presumably at terminal velocity, to the contact patch at $t = 86.67$ ms, merging with another droplet. In detail, it moves with an average velocity of about $\bar{v}_d = 0.26$ m $s^{-1}$ within the first 1.33 ms and about $\bar{v}_d = 6$ mm $s^{-1}$ between 1.33 ms and 86.67 ms, assuming the traveled distance $S_d$ equals the bubble radius $R_b = 509$ μm. The latter

correlates well with the terminal velocity of the droplet $v_t = 4.1$ mm $s^{-1}$ in the Stokes regime (Supplementary Note 4). The process repeats during numerous coalescence events, resulting in the gradual formation of electrolyte puddles, as shown in Fig. 3b. These puddles grow in size throughout the bubble evolution, as more electrolyte droplets are injected, wetting larger areas of the electrode. Once any of the puddles reaches the gas-electrolyte interface, it rapidly merges with the electrolyte bulk, thereby moving the contact line and effectively reducing the bubble-electrode contact area (see frames at 47.783 s and 64.650 s).

This process thus plays a key role for the bubble detachment. The detachment size of an electrode-attached bubble is primarily governed

by the surface tension force $F_s$, which depends on the length of the contact line ($L_c$). Comparing the snapshots at 47.783 s and 64.650 s, the length of the contact line can either increase or decrease after the puddle merges into the electrolyte bulk. A sudden reduction in $L_c$, provided there is sufficient buoyancy, causes an earlier detachment from the electrode surface, as illustrated in the snapshot at 64.667 s. The scarcity of electrolyte puddles in (d) is attributed to the reduced number of nucleation sites and their lower activity near the primary bubble, resulting in a lower frequency of coalescence events and fewer injected droplets.

## Worthington jet: electrolyte injection

Figure 4a shows a sequence of shadowgraphs detailing the mechanism of droplet injection characterized by the formation of an internal jet that entrains a volume of electrolyte, known in the fluid mechanics and physical oceanography communities as the Worthington jet[12]. The process is demonstrated by two coalescing $H_2$ bubbles with sizes $R_b = 400$ μm and $R_s = 205$ μm, respectively. The results are corroborated by direct numerical simulations (DNS) shown in Fig. 4b. In detail, when a smaller bubble touches a larger one, the liquid film that separates the bubbles gradually drains, forming a neck connecting the two ($t = 33.3$ μs)[45]. Growth of this neck follows a Taylor–Culick-type mechanism[45–47] exciting capillary waves that propagate along the bubble interface[48,49], see $t = 66.7$ μs –183.3 μs. Viscous forces damp the motion of these capillary waves, except for the strongest ones, which have the highest curvature. These waves concentrate at the bottom and induce a region of high curvature[13,50,51], see $t = 191.7$ μs–208.3 μs. This inertial flow focusing creates an upward jet ($t = 216.7$ μs–241.7 μs)[13,49,52] propagating inside of the merging $H_2$ bubbles. Consequently, this process is controlled by the dimensionless viscosity of the electrolyte given by the Ohnesorge number (Oh)

$$Oh = \frac{\mu_{el}}{\sqrt{\rho_{el}\gamma R_s}},$$ (1)

where $\mu_{el}$ represents the dynamic viscosity, $\rho_{el}$ the density of the electrolyte, $\gamma$ the surface tension, and $R_s$ the initial radius of the smaller bubble. Eventually, the jet breaks into two droplets due to the end-pinching mechanism[13,51], for Oh < Oh*. Here, Oh* ≈ 0.035 is the critical Ohnesorge number for the transition from drops ejection to no-drops ejection[13,51] for bursting of a jet at a free liquid-gas interface. The jet fragmentation results from the competition between the pressure-driven flow from the cylindrical jet toward the tip of the jet, and the capillary retraction of the jet tip. As a result of the latter, the tip is converted into a bulbous end, followed by a localized necking near the bulbous end, creating a blob. This blob at the jet tip detaches to form one droplet at a time. This pinching-off mechanism is consistent with the framework established previously for filament breakup[14–17]. Beyond the critical Ohnesorge number, viscous dissipation starts to dominate, ceasing the end-pinching of drops. We stress that in this regime, the jet could still break-up following the Rayleigh–Plateau mechanism[8,18]. On further increasing the Ohnesorge number (Oh > 0.1), the Worthington jet does not form, and there are no droplets[13,51].

The DNS results in Fig. 4b accurately reproduce key features and timescales of the phenomenon, such as neck formation, capillary wave propagation, formation, and breakup of the jet. In the experiments, the first droplet with a radius of $r_d = 13$ μm is observed at $t = 250.0$ μs and ejects with a velocity of approximately $\bar{v}_d = 7.2$ m s⁻¹. In close qualitative and quantitative agreement, the simulation demonstrates the first droplet (radius $r_d = 15$ μm) pinching off at $t = 260$ μs with $\bar{v}_d = 4.3$ m s⁻¹. Minor quantitative discrepancies arise from several factors. Experimental estimates of droplet size and velocity may be compromised due to the bubble curvature, which could be analytically corrected for a spherical shape. However, the significant deformations during the coalescence process complicate precise measurements. We stress that

the breakup process is a finite-time singularity[53]—so whether or not the jet breaks into droplets can be precisely reproduced in the simulations[18]—however, the time to break up and the velocity of the ejected drop are sensitive to the discrete nature of simulation method, experimental noise, and the measurement technique[45]. Additionally, the jet velocity, which determines the droplet post-ejection velocity, varies over time[13,48]; small variations in sampling time can thus explain discrepancies. Factors such as an inaccurate gas-liquid viscosity ratio and the sensitivity of velocity to the Ohnesorge number and size ratio also contribute to the differences between experimental and simulation results[51]. Given these error sources, the observed discrepancies are reasonable. Further details on the comparison between the experiments and simulations are provided in Supplementary Note 5.

It is important to note that the injection demonstrated at Oh = 0.008 in Fig. 4 represents a relatively isolated but conventional case[13,51], with the smaller bubble being located far from the electrode. In contrast, the bursting events in Figs. 1 and 2 taking place in a highly confined configuration near the Pt electrode feature high coalescence rates and involve smaller bubbles (up to about $R_s = \delta/2 = 8$ μm). Despite the higher Ohnesorge number (Oh = 0.042), injections still occur, exceeding the critical Oh* found for an unconfined isolated bubble. This observation suggests that a nearby wall and high coalescence rates can significantly influence the injection mechanism. In agreement with this, Lee et al. (2011)[54] identified a higher critical value Oh* = 0.052, specifically for smaller bubbles (with Bond number $Bo = \frac{\rho_{el}gR_s^2}{\gamma} < 10^{-3}$) bursting near a solid boundary. $g$ is the gravitational acceleration.

Lee et al. (2011)[54] also studied a bubble with a relatively small $R_s = 26.5$ μm adjacent to a Pt substrate using ultrafast X-ray imaging, finding daughter aerosol droplets (2 μm to 4 μm radii) with velocities around 0.3 m s⁻¹ (Supplementary Movie 5 in Lee et al.[54]). Consequently, we can classify the bursting events in order of increasing droplet speed: (i) carpet bubbles ($R_s = 8$ μm) bursting near a solid wall with a velocity of $v_d \sim 10^{-2}$ m s⁻¹, (ii) a bubble with $R_s = 26.5$ μm bursting near the solid wall, resulting in a droplet velocity of $v_d \sim 10^{-1}$ m s⁻¹, and (iii) the bubble with $R_s = 205$ μm bursting away from the wall (as detailed in Fig. 4), which results in a droplet velocity in the range of $v_d \sim 10^0$ m s⁻¹ to $10^1$ m s⁻¹. Further deceleration likely comes from viscous drag within the surrounding $H_2$ gas, as described by the Oseen approximation to the Stokes flow. Finally, a high coalescence rate, as seen in Figs. 1 and 2, could disrupt the symmetry of coalescence, affecting the propagation of capillary waves in each event and potentially significantly reducing the velocity of the ejected droplets to $v_d \sim 10^{-2}$ m s⁻¹. Therefore, the small initial size of the bursting bubble (i.e., large Oh), proximity to a wall, higher viscosity of the gas bubbles, and potentially high coalescence rates can substantially reduce the injection velocity.

## Discussion

Our findings demonstrate a distinct transport mechanism of electrolyte droplets inside the gas phase during water electrolysis. As discussed above, the coalescence of a primary bubble with the bubbles-satellites causes the electrolyte spraying via the fragmentation of the Worthington jet. This indicates that the $H_2$ bubble is not only composed of hydrogen gas and vapor but includes electrolyte fractions given the coalescence with nearby bubbles. We emphasize again that the microdroplets formed in the bubble through this process play an important role for the bubble detachment, once they merge with the surrounding electrolyte at the contact line. The results we report will be integral for further studying the limits of jet formation and rupture associated with Oh* in confined geometries near a solid boundary. Additionally, our findings will be valuable for validating and tailoring numerical and theoretical models. In particular, the droplet radii, injection speeds, and trajectories in various configurations, see the bubble-carpet system in Figs. 1 and 2, the surface-attached bubble system in Fig. 3 and the model experiment in Fig. 4, can serve as validation benchmarks. We highlight that the injected droplets serve as

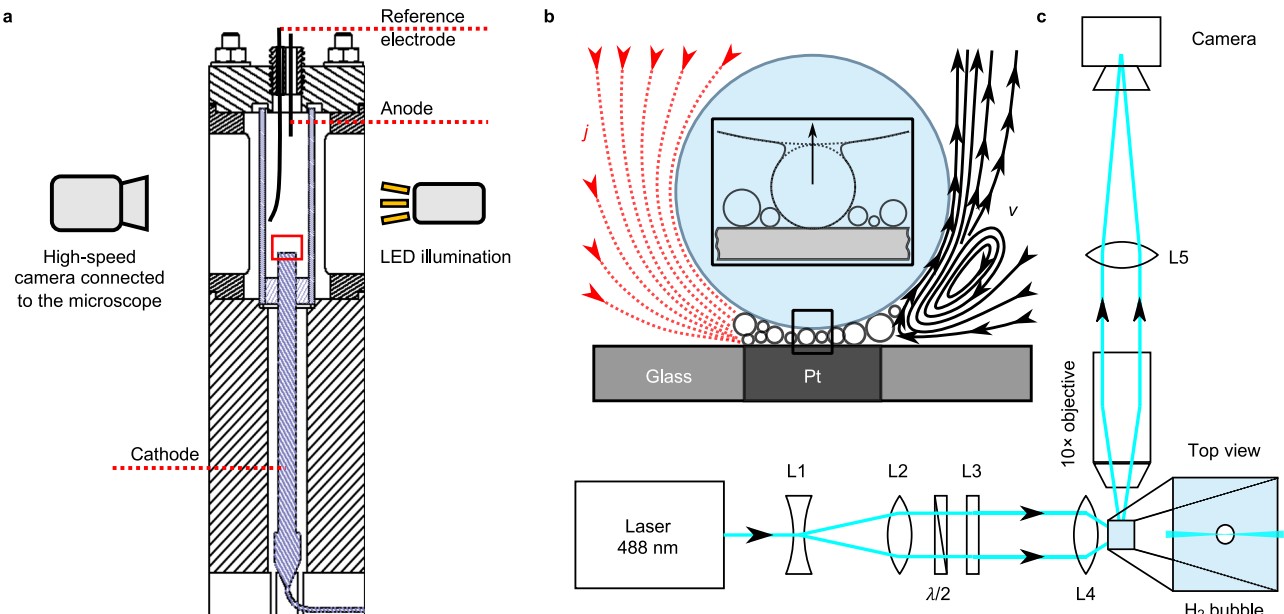

**Fig. 5 | Experimental schematics.** Schematics (not to scale) of (**a**) an electrochemical cell and a shadowgraphy system; (**b**) an $H_2$ bubble residing atop a carpet of microbubbles, generated between its bottom and the electrode surface, as shown by the red rectangle in (**a**). An inset zooms into the bottom of the bubble, marked by a black rectangle, where an intensive bubble-carpet coalescence takes place. The red lines represent the distribution of the current density ($j$), and the black streamlines on the right represent the Marangoni convection with velocity $v$. The panels (**a**) and (**b**) are adapted from Ref. [32]. (**c**) PTV optics used to measure the velocity of injected droplets inside the $H_2$ bubble. The cyan lines indicate the laser rays' path. For details, see text.

a non-invasive tool, making the internal flows associated with Marangoni convection at the electrolyte-gas interface visible and quantifiable. This gives access to the important surface mobility of electrogenerated bubbles, which is determined by preferential ion adsorption—a phenomenon that remains poorly understood. This will allow to assess the role of physicochemistry in the hydrodynamic phenomena related to bubbles. The knowledge could further be transferred to the other side of the electrochemical reaction—the formation of $O_2$ bubbles. The results of this work unravel important insights into the physicochemical aspects of electrochemically generated $H_2$ gas bubbles and have broad relevance, e.g. to acid mist formation in electrowinning processes[55]; the generation of sea spray aerosols[6], which play a role in airborne disease and pollutant transmission[21]; bursting $CO_2$ bubbles in sparkling drinks[5]; and to the impact of droplets on liquid[23] surfaces. In particular, the findings are essential for the water electrolysis field, where a deeper understanding of bubble evolution mechanisms is essential for optimizing gas-evolving electrochemical systems.

## Methods

The hydrogen gas bubbles were produced using both micro- and planar electrodes during water electrolysis. Part of the results (see Fig. 1) were obtained in a microgravity environment achieved during the 34th DLR Parabolic Flight Campaign in September 2019 (see Bashkatov et al.[41]).

### Materials

All chemicals were used without further purification process. The electrolytes were prepared from 1 mol L$^{-1}$ $H_2SO_4$ (Carl Roth GmbH & Co) and 1 mol L$^{-1}$ $HClO_4$ (Carl Roth GmbH & Co), and Milli-Q water ($\geq$18.2 MΩ cm).

### Microelectrode system

Single hydrogen gas bubbles growing on the carpet of microbubbles were produced using a three-electrode electrochemical cell filled with sulfuric acid of either 0.1 mol L$^{-1}$ or 0.5 mol L$^{-1}$ concentration, see

Fig. 5a. The cell used here closely resembles that used in earlier studies[27,56,57]. It comprises a cathode (Pt microelectrode, 99.99%, ⌀100 μm, ALS Co., Ltd) inserted horizontally facing upward in the base of a transparent cuboid glass cuvette (Hellma) with dimensions of $10 \times 10 \times 40$ mm$^3$, anode (⌀0.5 mm Pt wire, 99.99%, ALS Co., Ltd) and the Ag/AgCl reference electrode (BASi, Inc.) both inserted from the top. The experiments in a microgravity environment were done using a pseudo-reference electrode (identical to the anode)[41]. The electrochemical cell was fixed inside an outer housing featuring two optically accessible observation windows. Before the measurements, the microelectrode surface underwent mechanical polishing with sandpaper (2000 grit), sonication, and rinsing with ultrapure water ($\geq$18.2 MΩ cm). For microgravity experiments, it was polished by diamond (1 μm) and alumina (0.05 μm) suspensions (ALS Co., Ltd) instead. The cell was connected to an electrochemical workstation (CHI 660E, Autolab or Biologic VSP-300) and operated at a constant potential of either −3, −4, or −7 V.

The experiments using a blunt needle in Fig. 4 were performed as follows: A larger $H_2$ bubble (≈400 μm radius) was generated upfront at the microelectrode and detached upon potential interruption. As it ascended, it adhered to a blunt needle positioned above the microelectrode, with surface tension keeping the bubble attached. A second, smaller bubble (≈205 μm radius) was produced in the same manner, with the smaller size achieved by applying a shorter pulse of potential. As the smaller bubble rose, it contacted the larger bubble, initiating the coalescence process. The time $t_0$ marks the moment just before coalescence begins.

### Planar electrode system

The electrode-attached hydrogen gas bubbles were produced at the surface of a ⌀50 mm disc-like planar electrode (cathode) inserted horizontally facing upward in the base of the cylindrical PTFE (polytetrafluoroethylene) compartment with an inner ⌀ of 40 mm and a height of 50 mm filled with 0.1 mol L$^{-1}$ $HClO_4$. The cathode was fabricated by sputtering a 20 nm thin film of platinum onto a glass slide, with a 3 nm tantalum layer applied for improved adhesion. The thin layer of

platinum ensured the transparency of the cathode and allowed visualization from the bottom of the cell. The cell was completed by a platinized titanium mesh (anode) and the Ag/AgCl reference electrode (BASi, Inc.), both inserted from the top. The system was controlled by the electrochemical workstation (Biologic VSP-300) maintaining a constant current density of $-50\,A\,m^{-2}$. The relatively low current density and smooth surface of the cathode allowed only a limited number of active nucleation sites, making the study of the contact line and electrolyte puddles dynamics possible. For details, we refer to Demirkır et al.[34].

## Shadowgraphy system

The visualization of the bubble dynamics is performed using a conventional shadowgraphy system, shown schematically for a microelectrode system in Fig. 5a. It consists of a high-speed camera connected to the microscope and LED illumination. The shadowgraphs in Figs. 1 and 2a were recorded using an IDT camera (NX4-S1 and Os7-S3) with spatial resolutions of 678 and 1391 pixels $mm^{-1}$, respectively. In Figs. 3 and 4a, a Photron camera (FASTCAM NOVA S16) was used, with spatial resolutions of 530 and 496 pixels $mm^{-1}$, respectively. To achieve the bottom view (planar electrode system), the optical path of a horizontally installed camera is redirected vertically through the transparent cathode using a 45° mirror mounted below the electrode[34]. The LED light illuminates perpendicularly to the electrode from the top of the cell. The vertical adjustments of the focal plane are achieved using a high-precision motorized stage.

## Particle tracking velocimetry (PTV)

The evolution of $H_2$ bubbles at microelectrodes is featured by the intensive coalescence with the carpet of microbubbles sandwiched between the bubble bottom and electrode (Fig. 5b) on a time scale of μs. Owing to these coalescence events, multiple electrolyte droplets are injected into the bubble. The velocity measurement of these electrolyte droplets is performed using a particle tracking velocimetry (PTV) system, schematically shown in Fig. 5c.

The setup employs a light sheet optical configuration comprising a laser (OBIS 488LX, 150 mW, Coherent Inc.) that was spatially enlarged using a telescope (L1 & L2). To minimize reflection at the bubble surface, a $\lambda/2$-waveplate is employed to rotate the polarization. Subsequently, the beam is vertically expanded using a cylindrical lens (L3) before being focused inside the bubble by another lens (L4) with a focal length $f = 19\,mm$. For imaging purposes, a microscope objective (PLN 10× , Olympus) is positioned such that the bubble resides within the working distance. Finally, the bubble is imaged onto the camera (EoSens 3CXP, Mikrotron) using a lens (L5) with a focal length $f = 160\,mm$. To resolve the contours of the bubble, the system additionally possesses a background LED illumination. A series of images from Fig. 2b, c is collected at 1 kHz, having a spatial resolution of 1140 pixels $mm^{-1}$. Before velocity calculations, optical distortions (aberration) caused by the curvature of the bubble are analytically corrected (see next subsection). The resulting series of images were processed by the software DaViS 10, which employs a PTV algorithm to track each particle (droplet) over 25 ms at $t/T = 0.6$, 0.8 and 1.0. Due to the limited number of droplets, the resulting tracks were collected over several bubbles. Subsequently, the tracks were converted into a vector field using a binning function that interpolates local tracks on a specified fine grid. Finally, the vector fields are used to plot the streamlines of the averaged drop velocity field shown in Fig. 2.

## Analytical aberration correction

Aberrations (optical distortions) caused by light refraction at the curved gas-liquid interface of the bubble lead to a significant systematic measurement deviation[58,59]. As the bubbles are of submillimeter size, the correction using an optical element is challenging. However, for object-space telecentric lenses, like the microscope objective used for the measurements here, the aberrations can be

corrected analytically, which has already been done for the flows inside the droplets[60,61]. In the present study, the entrance pupil of the system is at infinity, resulting in chief rays parallel to the optical axis in the object space. When back-propagating the rays through the optical system, the intersection of the back-propagated ray and the light sheet (positioned in the middle of the bubble), can be calculated with and without the bubble in the system.

Supplementary Fig. 2a documents a schematic of the chief rays passing through the bubble surface. The solid lines mark the real path of the light scattered at the injected electrolyte droplets, whereas the dashed lines indicate the path recorded by the camera. Supplementary Fig. 2b demonstrates a single ray passing through the bubble-electrolyte interface with the relevant geometry used in further calculations of the corrected position for each detected droplet. Since the bubble is assumed to be axisymmetric, the position of the droplet is defined by the radial distance from the bubble center in the plane of the laser sheet. The measured and real (corrected) positions of the injected droplet are therefore defined as $r_{meas}$ and $r_{corr}$. The corrected position for each of the detected droplets can be calculated as $r_{corr} = r_{meas} + \Delta r$ using

$$\Delta r = \tan\left[\sin^{-1}\left[n \cdot \sin\left[\tan^{-1}\left(\frac{r_{meas}}{\sqrt{R_b^2 - r_{meas}^2}}\right)\right]\right] - \tan^{-1}\left(\frac{r_{meas}}{\sqrt{R_b^2 - r_{meas}^2}}\right)\right] \cdot \sqrt{R_b^2 - r_{meas}^2},$$

(2)

where $n$ is the refractive index of the electrolyte and $R_b$ is the radius of the bubble. The results of the analytical aberration correction and further details can be found in Supplementary Note 2 and Supplementary Fig. 3.

## Numerical method

In this work, the direct numerical simulation, using open source language Basilisk C, is employed to simulate the coalescence of two bubbles. A two-fluid model, combined with a Navier Stokes solver, is employed. The interface of the liquid and gas is tracked with the Volume of Fluid (VoF) method. The liquid phase is water with a density and dynamic viscosity of 1000 kg $m^{-3}$ and 0.00105 Pa s, respectively. The gas

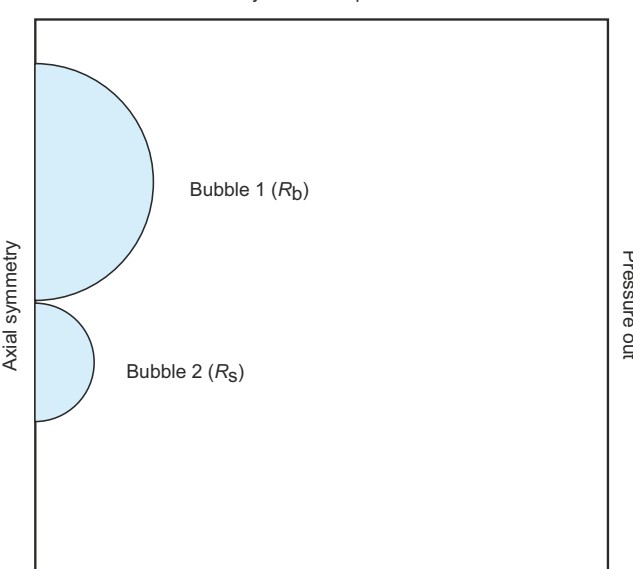

**Fig. 6 | Sketch of the simulation model.** Sketch of the simulation model illustrating two bubbles with radii $R_b$ and $R_s$ just before coalescence.

phase is air, with a density and dynamic viscosity of 1.41 kg m$^{-3}$ and 1.86 · 10$^{-5}$ Pa s. The surface tension on the interface of liquid and gas is 0.072 N m$^{-1}$. The initial radius of bubble 1 is $R_b$ = 400 μm and bubble 2 is $R_s$ = 200 μm. Figure 6 demonstrates a sketch of the simulation model.

Spatial discretization is performed using a quad-tree method in a 2D axisymmetric calculation domain of 1.5 · 10$^{-3}$ m × 1.5 · 10$^{-3}$ m. The adaptive Mesh Refinement algorithm was used to increase the calculation accuracy and reduce the hardware requirement. The maximum refinement level and the minimum level are 9 and 5, respectively. The calculation time step size is set to 1 · 10$^{-8}$ s. We emphasize that while present simulations assume axisymmetric bubbles as previous simulation work has done[13], real bursting events may demonstrate slight asymmetry, leading to the deviations in the jet/droplet bursting velocity and droplet radius.

## Data availability
The data that support the findings of this study are available from RODARE[62] and from the corresponding authors upon request. Source data are provided with this paper.

## Code availability
Codes used in this study are available from Zenodo[63] and from the corresponding author upon request.

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

## Acknowledgements
K.E., X.Y., and G.M. received funding from the German Space Agency (DLR), with funds provided by the Federal Ministry of Economics and Technology (BMWi) due to an enactment of the German Bundestag under Grant No. DLR 50WM2352 (project MADAGAS III), H2Giga (BMBF, 03HY123E), and from the Hydrogen Lab of the School of Engineering of TU Dresden. D.L. and D.K. acknowledge funding from the Advanced Research Center Chemical Building Blocks Consortium (ARC CBBC), under the project of New Chemistry for a Sustainable Future (project number 2021.038.C.UT.14). J.C. and L.B. acknowledge financial support from the German Research Foundation (DFG, project number 459505672). We thank Ayush K. Dixit for the discussions.

## Author contributions
A. Bashkatov and K.E. conceived the project. A. Bashkatov, F.B., Ç.D., A. Babich, X.Y., D.L., D.K., L.B., and K.E. designed the experiments. A. Bashkatov, F.B., and Ç.D. carried out the experiments. W.D. and V.S. carried out numerical simulations. A. Bashkatov, Ç.D., W.D., V.S., A. Babich, G.M., J.C., D.L., D.K., and K.E. carried out bubble dynamics analysis. All authors read and commented on the manuscript. All authors approved the final version of the manuscript.

## Funding

## Competing interests
The authors declare no competing interests.
