## [Transparent Peer Review File · Nature Communications]

Electrolyte droplet spraying within H₂ bubbles during water electrolysis under normal and microgravity conditions

Corresponding Author: Dr Aleksandr Bashkatov

Version 0:

Reviewer comments:

Reviewer #1

(Remarks to the Author)

In this article, a distinct transport mechanism whereby electrolyte droplets are sprayed into H₂ bubbles is evidenced, both experimentally and numerically. These droplets arise from the fragmentation of the so-called "Worthington jet", which is engendered by the coalescence with microbubbles.

Authors conclude that H₂ bubbles are therefore not only composed of hydrogen gas and water vapor but also include electrolyte fractions given the coalescence with nearby bubbles. Moreover, author evidenced that the microdroplets formed in the bubble through this process play an important role for the bubble detachment, once they merge with the surrounding electrolyte at the contact line.

Undoubtedly, this study has been properly carried out through high-speed imaging, but also concerning the digital modeling part of the article, thus bringing new insights into the physicochemical aspects of electrolytic gas bubbles.

I recommend publication of this article in Nat Communications, but I nevertheless highlight a little technical point to correct/clarify in the introduction about the mechanisms responsible for the fragmentation into multiple droplets of the so-called Worthington jet.

Indeed, to the best of my knowledge, the breakup of this very small jet is not a consequence of the Rayleigh-Plateau instability. Instead, the breakup takes place only at the jet tip, and detaches one droplet at a time. This mechanism, rather called "the end-pinching" mechanism, is a consequence of a competition between the capillary retraction of the jet tip (shaping a blob) and a pressure driven flow from the cylindrical jet toward the bulbous end (Keller et al, Phys Fluids (1995)). This leads to the development of a neck, clearly seen on the snapshots, where the jet joins the blob, and thus force the drop detachment via a capillary pinch-off process.

To the best of my knowledge, this mechanism has first been described in the context of a strongly deformed viscous drop (Stone et al, J. Fluid Mech (1989)), and later for a free liquid filament of arbitrary viscosity (Castejon-Pita et al, Phys Rev Lett (2012)).

Reviewer #2

(Remarks to the Author)

The work deals with the study of the formation of bubbles and the role of the electrolyte in the formation of these bubbles. I find the topic exciting and with a significant relevance in our understanding of the electrochemical systems. However, with this detailed data the support and development of computational methodologies to introduce all these aspects seems crucial to me. I understand that this might not be the area of expertise of the authors, but at least a discussion on the data generated or needed to inform the computational models could activate the work in this area.

Version 1:

Reviewer comments:

Reviewer #1

(Remarks to the Author)

My comments were carefully taken into account, I therefore recommend the publication of this paper in its present form

Reviewer #2

(Remarks to the Author)

The authors have successfully answered all the questions I raised to the previous version of the manuscript. Therefore, I warmly recommend the work for publication.

Response to the reviewers

We are very thankful to the reviewers for their critical reading of the manuscript and helpful comments and suggestions. All the changes incorporated in the manuscript are made in pink. Below, we have addressed the raised questions point-by-point:

Reviewer 1

In this article, a distinct transport mechanism whereby electrolyte droplets are sprayed into H_2 bubbles is evidenced, both experimentally and numerically. These droplets arise from the fragmentation of the so-called "Worthington jet", which is engendered by the coalescence with microbubbles. Authors conclude that H_2 bubbles are therefore not only composed of hydrogen gas and water vapor but also include electrolyte fractions given the coalescence with nearby bubbles. Moreover, author evidenced that the microdroplets formed in the bubble through this process play an important role for the bubble detachment, once they merge with the surrounding electrolyte at the contact line. Undoubtedly, this study has been properly carried out through high-speed imaging, but also concerning the digital modeling part of the article, thus bringing new insights into the physicochemical aspects of electrolytic gas bubbles. I recommend publication of this article in Nat Communications, but I nevertheless highlight a little technical point to correct/clarify in the introduction about the mechanisms responsible for the fragmentation into multiple droplets of the so-called Worthington jet.

Point 1.1 — Indeed, to the best of my knowledge, the breakup of this very small jet is not a consequence of the Rayleigh-Plateau instability. Instead, the breakup takes place only at the jet tip, and detaches one droplet at a time. This mechanisms, rather called "the end-pinching" mechanism, is a consequence of a competition between the capillary retraction of the jet tip (shaping a blob) and a pressure driven flow from the cylindrical jet toward the bulbous end (Keller et al. (1995)). This leads to the development of a neck, clearly seen on the snapshots, where the jet joins the blob, and thus force the drop detachment via a capillary pinch-off process. To the best of my knowledge, this mechanism has first been described in the context of a strongly deformed viscous drop (Stone and Leal (1989)), and later for a free liquid filament of arbitrary viscosity (Castrejón-Pita et al. (2012)).

Reply: We are very grateful to the reviewer for his/her excellent comment which has stimulated us to conduct new proof-of-concept simulations. Here we could show that the Worthington jet in our experiment indeed exhibits the end-pinching route, forecasted by the reviewer, rather than a Rayleigh-Plateau instability. The present pinching-off mechanism is consistent with the framework established previously for filament breakup (Stone and Leal, 1989; Keller et al., 1995; Castrejón-Pita et al., 2012; Anthony et al., 2019).

We have updated the text accordingly and cited the relevant references. We also note that depending on the viscosity ratio, jet geometry, and the timescale of bubble coalescence, other fragmentation pathways (e.g., Rayleigh-Plateau-like modes or film rupture mechanisms) could in principle operate in related setups such as coalescence of bulk bubbles as discussed recently in Jiang et al. (2024). In our present study, however, end-pinching is the dominant route for droplet formation within the confined bubble-carpet coalescence environment of the growing H_2 bubble. The main conclusion—that electrolyte droplets are driven inside the main H_2 bubble and eventually facilitate its detachment—remains

robust regardless of the fragmentation mechanism. We appreciate the reviewer’s suggestion and have emphasized that end-pinching is the primary mode in our case.

We incorporated the above discussion in the revised manuscript as follows:

In Introduction:

“... where the mechanism is either end-pinching (Stone and Leal (1989); Keller et al. (1995); Castrejón-Pita et al. (2012); Anthony et al. (2019)) or Rayleigh–Plateau instability (Blanco-Rodríguez and Gordillo (2020); Dixit et al. (2024)).”

In sec: Worthington jet: electrolyte injection:

“Eventually, the jet breaks into two droplets due to the end-pinching mechanism (Sanjay et al. (2021); Walls et al. (2015)), for $Oh < Oh^*$. Here, $Oh^* \approx 0.035$ is the critical Ohnesorge number for the transition from drops ejection to no-drops ejection (Sanjay et al. (2021); Walls et al. (2015)) for bursting of a jet at a free liquid-gas interface. The jet fragmentation results from the competition between the pressure-driven flow from the cylindrical jet toward the tip of the jet, and the capillary retraction of the jet tip. As a result of the latter, the tip is converted into a bulbous end, followed by a localized necking near the bulbous end, creating a blob. This blob at the jet tip detaches to form one droplet at a time. This pinching-off mechanism is consistent with the framework established previously for filament breakup (Stone and Leal, 1989; Keller et al., 1995; Castrejón-Pita et al., 2012; Anthony et al., 2019). Beyond the critical Ohnesorge number, viscous dissipation starts to dominate, ceasing the end-pinching of drops. We stress that in this regime the jet could still break-up following Rayleigh–Plateau mechanism (Blanco-Rodríguez and Gordillo (2020); Dixit et al. (2024)). On further increasing the Ohnesorge number ($Oh > 0.1$), the Worthington jet does not form and there are no droplets (Walls et al. (2015); Sanjay et al. (2021)).”

Reviewer 2

The work deals with the study of the formation of bubbles and the role of the electrolyte in the formation of these bubbles. I find the topic exciting and with a significant relevance in our understanding of the electrochemical systems.

Point 2.1 — However, with this detailed data the support and development of computational methodologies to introduce all these aspects seems crucial to me. I understand that this might not be the area of expertise of the authors, but at least a discussion on the data generated or needed to inform the computational models could activate the work in this area.

Reply:

We thank the reviewer for this comment. The main thrust of our article has been on experimentally uncovering the electrolyte-spraying mechanism, and its implications, in connection with the numerical simulations to reveal the underlying mechanism. However, our data indeed serve as a foundation for the computational community to incorporate these features into next-generation multiphase electrochemical flow solvers. In the revised manuscript, we highlight which particular datasets—such as droplet velocity fields, bubble interface topologies, and boundary conditions—are essential for refining multiphase electrochemical flow solvers.

Particularly, as computing power continues to improve significantly, direct numerical simulation (DNS) becomes viable for the research of fluid dynamics. With a well-suited adaptive mesh refinement method (Popinet, 2003, 2009, 2015; Beetham et al., 2016), the open-source partial differential equation solver Basilisk becomes well suited for resolving the multiphase flow problems such as bubble dynamics (Sanjay et al., 2021), bubble-related jetting (Deike et al., 2018; Yang et al., 2023) and droplet break-up (Brasz et al., 2018). Here, we use the open-source language Basilisk C to simulate the coalescence of two bubbles, the induced jet and the resulting droplet. The interface of the liquid and gas is tracked using the Volume of Fluid (VoF) method in the code. The liquid phase is water with a density and dynamic viscosity of 1000 kg/m^3 and $0.00105 \text{ Pa}\cdot\text{s}$, respectively. The gas phase is air, with a density and dynamic viscosity of $1.41 \text{ kg}\cdot\text{m}^{-3}$ and $1.46 \cdot 10^{-5} \text{ Pa}\cdot\text{s}$. The surface tension on the interface of liquid and gas is 0.072 N/m . The initial radii of bubble 1 and bubble 2 are $400 \mu\text{m}$ and $200 \mu\text{m}$, respectively. Figure 6 demonstrates a sketch of the simulation model. Spatial discretization is performed using a quad-tree method in a 2D axisymmetric calculation domain of $1.5 \cdot 10^{-3} \text{ m} \times 1.5 \cdot 10^{-3} \text{ m}$. The maximum refinement level and minimum level are set at 9 and 5 respectively based on our convergence tests and previous works (Yang et al., 2023). The calculation timestep size is set to $1 \cdot 10^{-8} \text{ s}$. The numerical simulations had a good agreement compared with experiments regarding the wave excitation, propagating, and jet/droplet formation as shown in Fig. 4. In addition, the present simulations assumed axisymmetric distributed bubbles as most previous simulation work has done (Sanjay et al., 2021), but jet/droplet bursting in reality might show slightly asymmetry which may offer a deviation on the jet/droplet bursting velocity and radius. In the spirit of open-source code development and reproducible open science, we have made the entire code base available on GitHub: <https://github.com/wding-hzdr/CoBub>.

We emphasize that capturing coalescence in real electrochemical environments will eventually require coupling potential fields, mass transfer of dissolved gas, and surface tension gradients arising from reactant depletion or thermal effects. Ongoing work seeks to incorporate these electrochemical boundary conditions, including local current densities and temperature fields, into multiphase solvers.

Our measurement data, in particular the droplet injection speeds and trajectories at different electrolyte properties, can serve as validation benchmarks.

To address the point raised by the reviewer and to further highlight the origin of the minor quantitative differences between simulations and experiments, we have incorporated the following revisions into the manuscript:

In sec: Worthington jet: electrolyte injection:

“Minor quantitative discrepancies arise from several factors. Experimental estimates of droplet size and velocity may be compromised due to the bubble curvature, which could be analytically corrected for a spherical shape. However, the significant deformations during the coalescence process complicate precise measurements. We stress that the breakup process is a finite-time singularity (Zeff et al. (2000)) – so whether or not the jet breaks into droplets can be precisely reproduced in the simulations (Dixit et al. (2024)) – however, the time to break up and the velocity of the ejected drop are sensitive to the discrete nature of simulation method, experimental noise, and measurement technique (Eggers et al. (2024)). Additionally, the jet velocity, which determines the droplet post-ejection velocity, varies over time (Deike et al. (2018); Sanjay et al. (2021)); small variations in sampling time can thus explain discrepancies. Factors such as an inaccurate gas-liquid viscosity ratio and the sensitivity of velocity to the Ohnesorge number and size ratio also contribute to the differences between experimental and simulation results (Walls et al. (2015)). Given these error sources, the observed discrepancies are reasonable. Further details on the comparison between the experiments and simulations are provided in Supplementary Section 5.”

In Conclusions:

“Additionally, our findings will be valuable for validating and tailoring numerical and theoretical models. In particular the droplet radii, injection speeds and trajectories in various configurations, see the bubble-carpet system in Fig. 1 and 2, the surface-attached bubble system in Fig. 3 and the model experiment in Fig. 4, can serve as validation benchmarks. ”

In Methods: Numerical method

“The calculation time step size is set to $1 \cdot 10^{-8}$ s. We emphasize that while present simulations assume axisymmetric bubbles as previous simulation work has done (Sanjay et al., 2021), real bursting events may demonstrate slight asymmetry leading to the deviations in the jet/droplet bursting velocity and droplet radius.”

References

- C. R. Anthony, P. M. Kamat, M. T. Harris, and O. A. Basaran. Dynamics of contracting filaments. *Phys. Rev. Fluids*, 4(9):093601, 2019.
- E. Beetham, P. S. Kench, J. O’Callaghan, and S. Popinet. Wave transformation and shoreline water level on f unafuti a toll, t uvalu. *Journal of Geophysical Research: Oceans*, 121(1):311–326, 2016.
- F. J. Blanco-Rodríguez and J. M. Gordillo. On the sea spray aerosol originated from bubble bursting jets. *J. Fluid Mech.*, 886:R2, 2020. doi: 10.1017/jfm.2019.1061.
- C. F. Brasz, A. Berny, and J. C. Bird. Threshold for discretely self-similar satellite drop formation from a retracting liquid cone. *Physical Review Fluids*, 3(10):104002, 2018.

- A. A. Castrejón-Pita, J. Castrejón-Pita, and I. Hutchings. Breakup of liquid filaments. *Physical review letters*, 108(7):074506, 2012.
- L. Deike, E. Ghabache, G. Liger-Belair, A. K. Das, S. Zaleski, S. Popinet, and T. Séon. Dynamics of jets produced by bursting bubbles. *Physical Review Fluids*, 3(1):013603, 2018.
- A. K. Dixit, A. Oratis, K. Zinelis, D. Lohse, and V. Sanjay. Viscoelastic worthington jets & droplets produced by bursting bubbles. *arXiv preprint arXiv:2408.05089*, 2024.
- J. Eggers, J. E. Sprittles, and J. H. Snoeijer. Coalescence dynamics. *Annual Review of Fluid Mechanics*, 57, 2024. doi: 10.1146/annurev-fluid-121021-044919.
- X. Jiang, L. Rotily, E. Villermaux, and X. Wang. Abyss aerosols: Drop production from underwater bubble collisions. *Phys. Rev. Lett.*, 133(2):024001, 2024.
- J. B. Keller, A. King, and L. Ting. Blob formation. *Phys. Fluids*, 7(1):226–228, 1995.
- S. Popinet. Gerris: a tree-based adaptive solver for the incompressible euler equations in complex geometries. *Journal of computational physics*, 190(2):572–600, 2003.
- S. Popinet. An accurate adaptive solver for surface-tension-driven interfacial flows. *Journal of Computational Physics*, 228(16):5838–5866, 2009.
- S. Popinet. A quadtree-adaptive multigrid solver for the serre–green–naghdi equations. *Journal of Computational Physics*, 302:336–358, 2015.
- V. Sanjay, D. Lohse, and M. Jalaal. Bursting bubble in a viscoplastic medium. *J. Fluid Mech.*, 922:A2, 2021. doi: 10.1017/jfm.2021.489.
- H. A. Stone and L. G. Leal. The influence of initial deformation on drop breakup in subcritical time-dependent flows at low reynolds numbers. *J. Fluid Mech.*, 206:223–263, 1989.
- P. L. Walls, L. Henaux, and J. C. Bird. Jet drops from bursting bubbles: How gravity and viscosity couple to inhibit droplet production. *Physical Review E*, 92(2):021002, 2015.
- Z. Yang, B. Ji, J. T. Ault, and J. Feng. Enhanced singular jet formation in oil-coated bubble bursting. *Nature Physics*, 19(6):884–890, 2023.
- B. W. Zeff, B. Kleber, J. Fineberg, and D. P. Lathrop. Singularity dynamics in curvature collapse and jet eruption on a fluid surface. *Nature*, 403(6768):401–404, 2000. doi: 10.1038/35000151.